# The Relevance of Optical Coherence Tomography Angiography in Screening and Monitoring Hypertensive Patients with Carotid Artery Stenosis

**DOI:** 10.3390/diagnostics15111393

**Published:** 2025-05-30

**Authors:** Irina Cristina Barca, Vasile Potop, Stefan Sorin Arama

**Affiliations:** 1Department of Physiology, “Carol Davila” University of Medicine and Pharmacy, 050474 Bucharest, Romania; 2Department of Ophthalmology, “Carol Davila” University of Medicine and Pharmacy, 050474 Bucharest, Romania

**Keywords:** optical coherence tomography angiography, carotid Doppler ultrasonography, retinal microvasculature, hypertensive retinopathy, screening, dyslipidemia

## Abstract

**Background**: Our study evaluated the correlation between internal carotid artery stenosis (ICAS) and retinal microvascular changes in patients with hypertensive retinopathy, dyslipidemia and ICAS. We analyzed vascular measurements provided by optical coherence tomography angiography (OCTA) and carotid Doppler ultrasonography (US) and linked OCTA parameters with carotid artery US measurements on the same side. Statistical differences in OCTA analysis among three groups (no stenosis, mild stenosis and moderate stenosis) were evaluated and correlated with carotid Doppler parameters. Our study aimed to evaluate whether OCTA can be proposed as a screening method in patients diagnosed with mild and moderate ICAS in order to improve the early detection of carotid changes, thus potentially reducing the rate of cardiovascular and cerebral complications of ICAS. **Methods**: We conducted a study on hypertensive patients with ICAS using six OCTA parameters in the analysis of the retinal vasculature and carotid Doppler US velocities of three carotid arteries and the vertebral artery (VA). Kruskal–Wallis and Dunn’s post hoc tests were used to determine whether there were statistically significant differences between the normal, mild and moderate stenosis groups. Spearman and Pearson correlation were used to obtain correlations among OCTA parameters such as the foveal avascular zone (FAZ), non-flow area (NFA), vascular flow area (VFA) and blood flow velocity on carotid Doppler US. **Results**: In the final analysis, 49 patients were included and 3 groups of stenosis were obtained, comprising 21 subjects with no stenosis, 19 with mild stenosis and 9 with moderate stenosis. Right eye and left eye groups were formed. In the right eye group with right ICAS, we found statistically significant results for FAZ circularity when comparing the normal stenosis group to the mild stenosis group (*p* = 0.025) and the mild stenosis group to the moderate stenosis group (*p* = 0.006). Statistically significant results were also observed for NFA when comparing the normal stenosis group to the moderate stenosis group (*p* = 0.004) and the mild stenosis group to the moderate stenosis group (*p* = 0.011). When comparing the FAZ area (*p* = 0.016) and VFA (*p* = 0.037) for the normal and moderate groups, statistically significant values were obtained. When comparing the normal and moderate stenosis groups with regard to the left eye, we found statistically significant results for VFA (*p* = 0.041), NFA (*p* = 0.045) and VFA (*p* = 0.029). When comparing the mild and moderate carotid artery stenosis groups, we obtained statistically significant results for NFA (*p* = 0.001), FAZ area (*p* = 0.007) and VFA (*p* = 0.013). In the right eye group, correlations between internal carotid artery (ICA) peak systolic velocity (PSV) and VFA (rho = −0.286), ICA end-diastolic velocity (EDV) and NFA (r = 0.365), external carotid artery (ECA) PSV and VFA (r = −0.288; rho = −0.317), common carotid artery (CCA) PSV and NFA (rho = −0.345), CCA EDV and NFA (rho = −0.292) and VA PSV and VFA (r = −0.327; rho = −0.379) were found. When analyzing OCTA parameters, we found statistically significant results for NFA and VFA (r = −0.374; rho = −0.288). Correlations were also found in the left eye group between ICA PSV and NFA (r = −0.351; rho = −0.313), ICA EDV and VFA (r = −0.421; rho = −0.314), ECA PSV and NFA (r = −0.412; rho = −0.457), CCA PSV and NFA (*p* = −0.288; rho = −0.339), and CCA EDV and NFA (r = −0.404; rho = −0.417). **Conclusions**: Our study found correlations between carotid Doppler velocities and OCTA vascular flow parameters; thus, OCTA may be used as a tool for monitoring the microvascular changes associated with carotid stenosis. OCTA can provide insights concerning the overall vascular condition of the patient, since it provides subjective data on vessel density and flow; therefore, by monitoring hypertensive patients with both OCTA and carotid Doppler US, we may be able to increase efficiency in screening and diagnosing patients with IACS.

## 1. Introduction

According to the World Health Organization, cardiovascular disease is the leading cause of death worldwide, representing the cause of 31% of the total deaths globally, with heart attack and stoke representing 85% of these casualties. An estimated 1.28 billion adults aged 30–79 years worldwide have hypertension. As one of the most important causes of ischemic cerebrovascular and eye diseases, carotid stenosis represents a public health burden globally. It is estimated that approximately 5–10% of the population over 70 years of age are affected by carotid stenosis [1,2].

By identifying cardiovascular risk factors such as hypertension, dyslipidemia, diabetes mellitus, smoking, obesity and alcoholic intake, we can prevent cardiovascular disease development and progression.

The pathophysiological changes implicated in the genesis of cardiovascular disease start developing well in advance of their diagnosis. Microvascular alterations are thoroughly associated with cardiovascular changes [2,3,4,5,6], so the early identification of patients at high risk is crucial.

The imaging of carotid plaques and the measurement of the internal carotid artery stenosis (ICAS) degree, based on the analysis of the peak systolic velocity and end-diastolic velocity, represent a simple, non-invasive and cost-effective method, and the reported overall accuracy of this method is up to 92% [7,8]. Applying carotid Doppler US screening to evaluate the ICAS degree in the general population could be a valuable tool to assist in cardiovascular risk stratification for the application of preventive strategies [8].

Though severe ICAS may lead to clinical visual signs and symptoms, patients with mild stenosis diagnosed using carotid Doppler US are usually asymptomatic. Vascular insufficiency, both in the optic nerve head and macula, may still lead to silent structural changes in the eye in the absence of symptoms.

The microvasculature of the retina is thought to have identical anatomic and physiological features to the coronary and cerebral microcirculation [3]. An increasing number of studies have indicated that retinal microvascular abnormalities may be used as a potential indicator for microvascular, cardiovascular and neurodegenerative disease [2,9,10,11].

The study of the retinal vasculature provides valuable information for observing microcirculation changes in patients with ICAS [2,3,5,12]. Blood vessels in the retina can be viewed directly, offering a readily accessible way to monitor vascular and circulatory function.

The ophthalmic artery is one of the branches of the internal carotid artery supplying blood to the eye. The central retinal artery arises from the ophthalmic artery. The choriocapillaris, superficial, and deep retinal layers are therefore all supplied by branches of the internal carotid artery (ICA). ICA narrowing can cause hypoperfusion of the retina and the choroid [13]. Given this anatomy, ICAS directly affects the ocular blood supply.

Several studies have demonstrated the association between ICAS and retinopathy, but the influence of mild ICAS on retinal vessels remains fully unknown [2,14,15]. Hence, identifying the changes in the retinal microvasculature in patients with mild forms of ICAS is of great significance to establish the disease status in a timely manner and to effectively monitor disease progression.

The retinal vasculature is one of the most easily visualized blood vessels in the body, providing an easy evaluation for monitoring systemic vascular conditions [2,3,5].

Several methods have been applied to evaluate ocular blood vessels, such as fundus photography and color Doppler imaging [2,16,17]. The velocity of blood flow in the ocular vessels, including the ophthalmic artery, the central retinal artery and the posterior ciliary artery can be effectively assessed using color Doppler imaging. The retinal vascular caliber can be measured using fundus photography to assess ocular blood flow in patients with ICAS [2,17]. However, given their inconvenient operation, low reliability and inability to measure the microvasculature of the retina, these methods cannot be widely used to observe retinal changes in ICAS patients.

Since it emerged, optical coherence tomography angiography (OCTA) has changed the process of quantifying and visualizing the retinal microvasculature, as many studies show [2,18,19,20]. OCTA is a dye-free, non-invasive and efficient method for the qualitative and quantitative assessment of retinal vessels [1,2,21,22].

OCTA is useful as it may reveal microvascular abnormalities before they become clinically apparent. Macular and peripapillary vessel density reduction has previously been reported in patients with cardiovascular diseases [2,8].

It has been demonstrated that the risk of cardiovascular disease can be monitored by quantifying retinal microvascular alterations using OCTA [2,3,20]. In addition, recent studies have demonstrated that the optic nerve and macular flow densities are significantly lower in moderate or severe ICAS patients [1,2,23,24,25]. These findings suggest that OCTA may be a useful tool for monitoring ICAS. Hence, the research on retinal microvasculature for different degrees of ICAS using OCTA can be improved.

Our research aimed to quantify and compare retinal microvasculature changes found through OCTA in patients with different degrees of ICAS and to correlate OCTA data with carotid Doppler US results. We measured and compared the retinal microvascular parameters provided by OCTA. We determined the correlations among OCTA parameters and carotid Doppler ultrasonography (US) measurements.

We hypothesized that retinal vasculature changes are related to mild carotid artery stenosis and moderate stenosis, as determined by carotid Doppler US screening. Therefore, OCTA screening can be performed on the same timeline as US in patients with mild and moderate stenosis.

The retinal microvascular changes assessed by OCTA could be potential biomarkers for identifying ICAS. After demonstrating diminished retinal microvascular density using OCTA in patients with Fabry disease, M. Rinaldi et al. concluded in their 2024 study that identifying biomarkers for assessing ocular vascular involvement and treatment response was imperative and that further research was needed [26].

A biomarker is defined as “a characteristic that is objectively measured and evaluated as an indicator of normal biological processes, pathogenic processes, or pharmacological responses to a therapeutic intervention” [23,27]. A biomarker useful in the prevention of IACS progression would (1) non-invasively detect early preclinical disease before the first clinical signs appear, (2) be causally linked or be an indicator of a causal mechanism that leads to the development of IACS and (3) be consistently and strongly associated with IACS. Therefore, our research also aimed to seek if biomarkers can be obtained.

By extending US screening and using data obtained in OCTA scans, we can assume that both examinations can be used as a screening method.

## 2. Materials and Methods

### 2.1. Study Population

The study was conducted according to the tenets of the Declaration of Helsinki.

Informed consent was obtained from all patients. The study was approved by the Ethics Committee of the “Regina Maria: The Healthcare Network” (645 bis/14.12.2022).

This prospective cohort study was performed in Eastern Europe, specifically Romania, from 2022 to 2024. We included 67 patients with dyslipidemia and systemic hypertension. Eighteen patients were excluded due to poor-quality images of the OCTA scans. Ninety-eight eyes were eligible for the final analysis. All participants underwent comprehensive ophthalmologic examination, carotid Doppler ultrasonography and evaluation of cardiovascular risk factors according to protocol.

In an attempt to respond to the question “Which retinal microvasculature changes can we find through OCTA in patients with different degrees of Internal carotid Artery stenosis (ICAS)?”, we conducted research using OCTA analysis and carotid Doppler US measurements.

For the purpose of this study, we selected patients diagnosed with chronic hypertension, secondary dyslipidemia and chronic hypertensive retinopathy. Participants with diabetes mellitus were excluded. The aim of our research was to enroll subjects irrespective of their current treatment for hypertension and dyslipidemia in order to study the correlation between hypertensive retinopathy and ICAS for all patients. We did not aim to study the effect of medication on the microvasculature of the retina.

Based on fundoscopy, we included in this study subjects with grade I and II hypertensive retinopathy according to the Keith–Wagener–Barker and the Mitchell–Wong classification systems [28]. Our research did not find participants with grade III and IV hypertensive retinopathy. Based on carotid Doppler US evaluation and on Society of Radiologists in Ultrasound consensus, patients were classified and all degrees of ICAS were included in our study.

Participants with the following pathologies were excluded with the help of fundoscopy, optic nerve OCT and retina OCTA: diabetic retinopathy, glaucoma, macular telangiectasia, age-related macular degeneration, central serous retinopathy, pigment epithelial detachments, epiretinal membranes, cystoid macula edema, vitreomacular interface disorders and other pathologies affecting retinal vasculature.

Eyes with ICAS-related complications such as ocular ischemic syndrome, central or branch retinal artery occlusion and anterior ischemic optic neuropathy were also excluded. Patients with heart failure, aortic stenosis or aneurysm, any other obstruction of the proximal vascular system from internal carotid artery (ICA), cerebrovascular accidents, neurodegenerative diseases, hematologic diseases and malignancies were excluded.

We performed measurements according to protocol: ophthalmological examinations and carotid Doppler US were performed every 3 months.

According to the American Heart Association criteria, hypertension is defined as systolic blood pressure ≥ 130 mm Hg and diastolic blood pressure ≥ 80 mm Hg [29,30,31]. A blood pressure higher than 180/120 mm Hg is considered a hypertensive emergency or crisis. Our research excluded subjects with a hypertensive emergency.

Participants included in our study were diagnosed with dyslipidemia prior to our study period. Smoking habit was assessed with the single categorized question “Are you currently a smoker?”. The age and sex were also noted.

Glycemia and glycated hemoglobin (HbA1c) were determined in order to rule out subjects with diabetes mellitus and diabetic retinopathy.

Each participant underwent an extensive ophthalmic examination including best corrected visual acuity (Snellen Chart), slit-lamp examination (Topcon, SL-D701), refraction (Auto Kerato-Refractometer Topcon, KR-800) and measurement of the intraocular pressure (Non-contact Tonometer, NT-530/510). Subjects with high myopia were excluded because of the optical aberration on OCT measurements. Patients with dense cataracts were excluded as visualization of the retina was impaired. Participants with an intraocular pressure below 10 mmHg or above 21 mmHg were excluded from our study. Fundoscopy was performed using a Volk 90D power lens in order to stage hypertensive retinopathy. Optopol Revo NX 130 OCTA of the macula was performed on both eyes of all patients every 3 months, with the assessment of the foveal avascular zone (FAZ), the non-flow area (NFA), the vascular flow area (VFA), retinal vascular density and perfusion. Eyes with poor-quality images or poor signal strength were excluded.

All patients were subjected to carotid Doppler ultrasonography using G.E. Vivid T8 Cardiovascular Ultrasound. The Doppler ultrasound is a valuable technique in the classification of carotid obstruction. The degree of stenosis was evaluated using the grading method suggested by the Society of Radiologists at the Ultrasound Consensus Conference. Stenosis was classified into multiple categories: normal (no stenosis), mild (<50% stenosis), moderate (50–69% stenosis), severe (≥70% stenosis but less than near occlusion), near occlusion and total occlusion of the ICA [9]. Carotid stenosis of more than 50% was considered hemodynamically significant.

Participants were divided into two groups: the right eye and the left eye group. Subjects were also divided into three groups regarding stenosis: without ICAS (normal group with 21 participants), mild ICAS (mild group with 19 participants) and moderate ICAS (moderate group with 3 participants). Our study did not find a severe ICAS group (0 participants) according to the grading method suggested by the Society of Radiologists. OCTA measurements were performed on both eyes in patients to determine the presence of unilateral or bilateral stenosis.

### 2.2. Examination and OCTA Analysis

OCTA of the macula was performed using Angio-OCT Optopol Revo NX 130 with AI technology: iTracking™ technology, Motion Correction Technology™ and the Projection Artefact Removal algorithm. A.I.DeNoise™ technology improves tomogram quality powered by AI so that the highest and smoothest image quality is obtained. The Maximum Intensity Projection (MIP) tool is useful for visualizing OCTA data, as it enables the easier identification and tracking of high-intensity structures such as blood vessels.

Eyes with significant artifacts that obscured the vascular area of interest, with segmentation failure that could not be corrected manually or by using iTracking™ technology and with poor-quality images or a poor signal strength, were excluded.

En face angiograms of the superficial capillary plexus, superficial vascular plexus, radial peripapillary capillary plexus, deep capillary plexus, deep vascular plexus, intermediate capillary plexus, outer retina layers, choriocapillaris and choroidal vessels with the depth coded were generated by automatic segmentation to assess the retinal microvascular perfusion.

For the purpose of our study, we performed 6.0 × 6.0 mm scans centered on the fovea of both eyes. The software (Version 11.0.4 REVO NX Device SN: 1560658/16) was used to analyze the superficial layer (offset of the inner limiting membrane (ILM)—0 µm; inner plexiform layer/inner nuclear layer (IPL/INL)—15 µm), the deep layer (offset of the IPL/INL—15 µm; IPL/INL—70 µm) and the choriocapillaris (offset of Bruch’s membrane (BM): top BM—30 µm; bottom BM—45 µm).

The OCTA provided assessment of the FAZ, NFA, VFA, the retinal vascular density and the skeleton. The FAZ, NFA and VFA were detected and measured using the semi-auto area tool of the software. In a number of scans where the semi-automated measurement proved to be inaccurate, manual measurements were performed.

Density and skeleton displays can be used to detect abnormal vasculature and provide repeatable quantitative results. The measurement zones available for retina scans with a width of 6 mm are total, superior, inferior, center, inner, superior inner, inferior inner, outer, superior outer, inferior outer and the Early Treatment of Diabetic Retinopathy Study (ETDRS) grid.

A comparison view screen was used to observe follow-up changes in the eye structure by comparing two examinations of the same eye. A progression view screen was used to show the analysis results comparing a maximum of four examinations, performed on the same side, in the same scan mode and on the same size of scanning area, arranged in a time sequence.

Figure 1, Figure 2 and Figure 3 display OCTA parameters and view screens.

### 2.3. Statistical Analysis

All analyses were performed using IBM Statistical Package for the Social Sciences (SPSS) V26, Minitab V20.3, JASP V0.19.3 and Microsoft Office Professional Plus 2021.

The whole sample of 98 eyes was divided into the right eye and left eye groups, with 49 patients each. Participants were also divided into 3 groups regarding stenosis: the normal group, with 21 subjects, the mild stenosis group, with 19 subjects, and the moderate stenosis group, with 9 subjects.

Using descriptive statistics, the following indicators were calculated: mean and standard deviation. We also analyzed age, sex and smoking habits.

The Kruskal–Wallis test was used to determine if there were statistically significant differences between the normal, mild and moderate stenosis groups. To compare different stenosis groups, we also used Dunn’s post hoc test following the Kruskal–Wallis test.

Spearman and Pearson correlation were used to obtain the correlations among right eye and left eye OCTA parameters and the peak systolic velocity (PSV) and end-diastolic velocity (EDV) calculated for the internal carotid artery (ICA), the external carotid artery (ECA), the common carotid artery (CCA) and the vertebral artery (VA).

The normality of the distribution was calculated using the Shapiro–Wilk test for multivariate normality, charts and quantile–quantile (QQ) plots.

Each QQ plot displays the distribution of data, regression line, correlation type and confidence interval in the right eye and left eye groups and the carotid arteries on the same side.

## 3. Results

### 3.1. Descriptive Analysis/Study Population

A total of 49 subjects with ICAS were enrolled in our research. Twenty-one patients had no stenosis according to the Society of Radiologists grading, nineteen patients had mild stenosis and a total of nine subjects had moderate stenosis. We did not find any patients with severe stenosis in our research.

A chi-square (ꭓ2) test and a Kruskal–Wallis test were used to analyze gender, smoking status and age in subjects with mild, moderate and no stenosis. The mean age of the patient group was 60.9 years with a SD of ±8.2 in the mild stenosis group, 54.4 ± 10.0 in the normal group and 58.2 ± 13.1 in the moderate stenosis group. The majority of subjects were males, representing a proportion of 63.2% in the mild group and 52.4% in the normal group. Non-smokers represented the majority in the normal group (80.9%) and the mild stenosis group (63.2%). In the moderate stenosis group, 88.9% of subjects were smokers (*p* = 0.002). The results are displayed in Table 1.

Table 1 displays the results of demographic analysis, including gender, smoking habit and age analysis. Data are displayed according to the degree of stenosis: normal, mild and moderate.

### 3.2. Analysis of OCTA Parameters in Patients with Carotid Stenosis

#### 3.2.1. Analysis of Right Eye OCTA Parameters

The Kruskal–Wallis test was used to determine if there were statistically significant differences between stenosis groups. To compare different groups, we also used Dunn’s post hoc test following Kruskal–Wallis test in both eyes.

Table 2 displays the mean deviation, standard deviation and comparisons between carotid artery stenosis groups when analyzing the right eye. We also compared stenosis groups, namely by comparing the normal and moderate groups, the normal and mild groups, and the mild and moderate groups.

The OCTA software provides assessment of the FAZ, NFA and VFA; the retinal vascular density; and the skeleton. Our research included only FAZ, NFA and VFA. These areas were detected and measured using the semi-auto area tool of the software, and thus the results proved to be more objective and accurate. The AI system of the OCTA software has a direct impact on the accuracy of the clinical assessment and the assessment of the status of areas of pathology in the retina. This level of detection is more accurate, and this results in more detailed screening. AI segmentation is important for follow-up examinations, providing a more accurate diagnosis when analyzing pathology over time.

Our research aimed to study if vascular changes at the level of three OCTA parameters are expected in patients with ICAS and to correlate carotid Doppler velocities with OCTA parameters.

When analyzing the right eye group, we found statistically significant results for FAZ circularity when comparing the normal and the mild stenosis groups (*p* = 0.025) and when comparing the mild and the moderate stenosis groups (*p* = 0.006). Correlations were also obtained for NFA when comparing the normal and moderate groups (*p* = 0.004) and the mild and moderate groups (*p* = 0.011). When comparing the normal and moderate groups, FAZ area (*p* = 0.016) and VFA (*p* = 0.037) statistically significant correlations were observed.

Figure 4, Figure 5, Figure 6, Figure 7 and Figure 8 display OCTA parameters’ distribution for each carotid artery stenosis group when analyzing the right eye and a boxplot for all three stenosis groups.

Figure 5, Figure 6, Figure 7 and Figure 8 display the distribution for each OCTA parameter as a scatter plot, boxplot and normal curve.

#### 3.2.2. Analysis of Left Eye OCTA Parameters

Table 3 displays the mean deviation, standard deviation and comparisons between carotid artery stenosis groups when analyzing the left eye. We also compared stenosis groups, namely by comparing the normal and moderate groups, the normal and mild groups, and the mild and moderate groups.

Dunn’s post hoc test was used following the Kruskal–Wallis test.

When comparing the normal and the moderate group in the left eye, we found statistically significant results for VFA (*p* = 0.041), NFA (*p* = 0.045) and VFA (*p* = 0.029). When comparing the mild and moderate carotid artery stenosis groups, we obtained statistically significant results for NFA (*p* = 0.001), FAZ area (*p* = 0.007) and VFA (*p* = 0.013).

Figure 9, Figure 10, Figure 11, Figure 12 and Figure 13 display the OCTA parameter distribution for each carotid artery stenosis group when analyzing the left eye. Scatter plots, boxplots and normal curves are represented.

#### 3.2.3. OCTA Parameters and Correlations for Both Eyes

We used Spearman and Pearson correlation to obtain correlations among right eye and left eye OCTA parameters and the PSV and EDV calculated for ICA, ECA, CCA and VA on both carotid arteries. The results are shown in Table 4 and Table 5.

By using Pearson correlation in the right eye group, statistically significant results were found for ICA PSV and VFA (rho = −0.286), ICA EDV and NFA (r = 0.365), ECA PSV and VFA (r = −0.288; rho = −0.317), CCA PSV and NFA (rho = −0.345), CCA EDV and NFA (rho = −0.292), VA PSV and VFA (r = −0.327; rho = −0.379). When analyzing OCTA parameters, we found statistically significant results for NFA and VFA (r = −0.374; rho = −0.288).

According to Pearson’s and Spearman’s rank correlation, the strongest correlations in the left eye group were found for ICA PSV and NFA (r = −0.351; rho = −0.313), ICA EDV and VFA (r = −0.421; rho = −0.314), ECA PSV and NFA (r = −0.412; rho = −0.457), CCA PSV and NFA (*p* = −0.288; rho = −0.339), and CCA EDV and NFA flow area (r = −0.404; rho = −0.417).

Normal distribution was calculated using the Shapiro–Wilk test for multivariate normality, charts and QQ plots. Each right eye QQ plot displays the distribution of data, regression line, correlation type and confidence interval.

Test results of the Shapiro–Wilk test for multivariate normality are displayed in Table 6 and Table 7 for the right eye group and the left eye group, respectively.

The test results of the Shapiro–Wilk test for multivariate normality for the left eye group demonstrated a significance level *p* < 0.001, and thus these data have a normal distribution.

The test results of the QQ plots in both eyes analyzing OCTA parameters and carotid Doppler velocities on the same side are available in the Appendix A.

The test results of the QQ plots analyzing OCTA parameters are displayed in Figure 14 and Figure 15.

## 4. Discussion

Many studies have applied OCTA in the study of systemic cardiovascular disease.

An increasing number of studies have indicated that retinal microvascular abnormalities may be used as a potential indicator for microvascular, cardiovascular and neurodegenerative disease [2,9,11].

In 2023, Haipeng Liu et al. presented the cardiocerebral arterial network based on the classic brain circulation model proposed by Alastruey et al. in 2007 [32]. The structure of the artery network starts from the aorta and includes the major branches of intracranial arteries. The intermediate branches included the ascending aorta, aortic arch (in two segments), brachiocephalic artery, common carotid arteries (left and right), subclavian arteries (left and right), vertebral arteries (left and right), internal carotid arteries (left and right, both in two segments), basilar artery, as well as the connecting arteries in the Circle of Willis, i.e., posterior communicating arteries (left and right), anterior communicating artery and the first segments of anterior and posterior cerebral arteries (left and right, for both). The extracranial arteries included the thoracic aorta, brachial arteries (left and right) and external carotid arteries (left and right) [33].

According to Arnould L. et al. in a study conducted in 2018, in most patients, the underlying pathophysiological process of cardiovascular disease develops long before their diagnosis. Simple examinations to detect early vascular remodeling would be useful for early identification of high-risk subjects.

Previous studies like the ones developed by Cheung N. et al. have suggested that microcirculatory changes are closely related to cardiovascular modifications in humans [34,35].

The ophthalmic artery is one of the branches of the internal carotid artery supplying blood to the eye. The central retinal artery arises from the ophthalmic artery. The choriocapillaris, superficial, and deep retinal layers are therefore all supplied by branches of the internal carotid artery. ICA narrowing can cause hypoperfusion of the retina and the choroid [13]. Given this anatomy, ICAS directly affects the ocular blood supply.

The study of the retinal vasculature provides valuable information for observing microcirculation changes in patients with ICAS [2,3,5,12]. Blood vessels in the retina can be viewed directly, offering a readily accessible way to monitor vascular and circulatory function. Microvascular abnormalities are demonstrated to be important contributors to cardiovascular disease [2,5,6].

ICAS is a major risk factor for ophthalmic ischemic complications such as ocular ischemic syndrome, amaurosis fugax and central retinal artery occlusion. Before ocular findings, fundus changes and visual symptoms, imaging of the retinal vascular structure may be an earlier marker for ICAS. Retinal vascular changes are usually asymptomatic and occur in up to 29% of patients with carotid artery occlusion, and 1.5% of cases progress to symptomatic ocular ischemic syndrome each year [36,37]. Hence, the early detection of retinal ischemia is vital for patients with ICAS diagnostic.

In a more recent study conducted in 2024 by Rinaldi M et al., the impact of Fabry disease on retinal microvasculature was evaluated using OCTA. Arterial stiffness and the resistive index of the central retinal artery in the early disease stages were calculated. Fabry disease patients demonstrate elevated arterial stiffness, diminished retinal microvascular density compared to healthy controls and an increased resistive index of the central retinal artery, indicating early ocular damage. According to their study, identifying biomarkers for assessing ocular vascular involvement and treatment response is imperative and further research is needed [26].

In 2022, Chiosi F et al.’s study [38] aimed to determine whether vessel density as measured by OCTA provides insights into retinal and choriocapillaris vascular changes in patients with SARS-CoV-2 infection. Their results demonstrated that foveal vessel density at the level of the deep capillary plexus and the choriocapillaris was significantly lower in post-COVID-19 patients when compared to healthy individuals.

The research on the retinal microvasculature for different systemic cardiovascular diseases, including the degree of ICAS, using OCTA can be improved.

ICAS is a worldwide public health issue. This carotid pathology causes ocular and cerebral ischemic events, which could lead to permanent disability or even death [39]. ICAS is a major risk factor for ophthalmic ischemic complications such as amaurosis fugax, ocular ischemic syndrome and central retinal artery occlusion. ICA narrowing can cause hypoperfusion of the retina and the choroid [13].

The microvasculature of the retina is thought to have identical anatomic and physiological features to the coronary and cerebral microcirculation [3]; therefore, the small blood vessels in the retina can provide useful information in patients with cerebral and cardiovascular risk factors.

Several studies have demonstrated the association between internal carotid artery stenosis and retinopathy, but the influence of mild ICAS on retinal vessels remains fully unknown [2,14,15]. Hence, by identifying the changes in retinal microvasculature in patients with mild forms of ICAS, we might be able to accurately evaluate the disease status and to effectively monitor disease progression.

Macular vessel density reduction was previously reported in patients with cardiovascular diseases [2,8]. Sayin et al. observed a decreased choroidal thickness in patients with ICAS [40].

It has been demonstrated that the risk of cardiovascular disease can be monitored by quantifying retinal microvascular alterations using OCTA [2,3,20]. In addition, recent studies have demonstrated that macular flow densities are significantly lower in moderate or severe ICAS patients [1,2,23,24,25]. These findings suggest that OCTA may be a useful tool for monitoring ICAS.

The American Heart Association 2013 guidelines recommend that it is reasonable to repeat carotid Doppler US screening annually for patients with moderate carotid artery stenosis, but not for those with mild stenosis [8,41,42].

Our study aimed to test a hypothesis and to answer a question regarding microvascular changes in patients with ICAS.

The purpose of this study was to quantify and non-invasively assess the retinal vascular and structural changes in patients with various degrees of ICAS by using OCTA. We measured and compared the retinal microvascular parameters provided by the OCTA. We determined correlations among OCTA parameters and carotid Doppler ultrasonography measurements on the same side.

We hypothesized that retinal vasculature changes in one eye are related to artery changes in subjects with mild and moderate carotid artery stenosis on the same side as determined by carotid Doppler US screening; thus, OCTA screening can be performed on the same timeline as US.

The whole sample of 98 eyes was divided into the right eye and left eye groups, with 49 patients each. We obtained 3 groups regarding stenosis, namely the normal group, with 21 subjects, the mild stenosis group, with 19 subjects, and the moderate stenosis group, with 9 subjects, and we analyzed data by using descriptive statistics.

The Kruskal–Wallis test and Dunn’s post hoc test were applied to determine whether there were statistically significant differences between the normal, mild and moderate stenosis groups.

Spearman and Pearson correlation were used to obtain correlations among right eye and left eye OCTA parameters and the PSV and EDV calculated for ICA, ECA, CCA and VA.

Normal distribution was calculated with the Shapiro–Wilk test for multivariate normality, charts and quantile-quantile (QQ) plots.

In patients with right ICAS, we found statistically significant results for FAZ circularity when comparing the normal group and the mild stenosis group (*p* = 0.025) and when comparing the mild and moderate stenosis groups (*p* = 0.006). Correlations were also obtained for NFA when comparing the normal and moderate groups (*p* = 0.004) and the mild and moderate groups (*p* = 0.011). When comparing the normal and moderate groups, statistically significant values were obtained for FAZ area (*p* = 0.016) and VFA (*p* = 0.037).

In the left eye ICAS group, when comparing the normal and moderate stenosis groups, we found statistically significant results for VFA (*p* = 0.041), NFA (*p* = 0.045) and VFA (*p* = 0.029). When comparing the mild and moderate carotid artery stenosis groups, we obtained statistically significant results for NFA (*p* = 0.001), FAZ area (*p* = 0.007) and VFA (*p* = 0.013).

By using Pearson correlation in the right eye group, statistically significant results were found for ICA PSV and VFA (rho = −0.286), ICA EDV and NFA (r = 0.365), ECA PSV and VFA (r = −0.288; rho = −0.317), CCA PSV and NFA (rho = −0.345), CCA EDV and NFA (rho = −0.292) and VA PSV and VFA (r = −0.327; rho = −0.379). When analyzing OCTA parameters, we found statistically significant results for NFA and VFA (r = −0.374; rho = −0.288).

According to Pearson’s and Spearman’s rank correlation, the strongest correlations in the left eye group were found for ICA PSV and NFA (r = −0.351; rho = −0.313), ICA EDV and VFA (r = −0.421; rho = −0.314), ECA PSV and NFA (r = −0.412; rho = −0.457), CCA PSV and NFA (*p* = −0.288; rho = −0.339) and for CCA EDV and NFA (r = −0.404; rho = −0.417).

Thus, by analyzing OCTA parameters and internal carotid artery US on the same side, our research provided useful data in order to propose that both US and OCTA screening be performed in hypertensive patients on the same timeline.

The study performed in 2024 by Jinlan Ma, NanJia Gelie, Mingjuan Zhu, Xiaolu Ma and Changjing Han quantified ocular microcirculation in hypertension patients with ICAS. Compared to our study, their study included OCTA metrics like vessel density, skeleton density, fractal dimension and the analysis of EDI-OCT. Choroidal microcirculation metrics, including subfoveal choroidal thickness, luminal area and choroidal vascularity index, were quantified using EDI-OCT. Their study also included a healthy control group. FAZ was analyzed in both studies as a sign of foveal ischemia. No significant differences in FAZ among the three groups (healthy control, hypertension with ICAS and hypertension with no ICAS) were found [43]. Our study did not include a healthy group, but also did not find significant differences in FAZ among our three stenosis groups (normal, mild and moderate).

The identification of subclinical carotid artery stenosis during the asymptomatic phase is an important public health and economic goal [44]. OCTA may be a valuable tool in detecting early microvascular changes in the retina, even before they become clinically apparent. The microvascular changes in the retina may also be used as an indicator of cardiovascular diseases [5,10].

It is important for clinicians to be familiar with ophthalmic symptoms and signs caused by carotid atherosclerosis in order to provide an early diagnosis and to take appropriate measures to prevent a stroke. However, even before ocular findings, fundus changes and visual symptoms, imaging of the retinal vascular structure may be an earlier marker for ICAS.

By identifying early microvascular improvements via OCTA scanning after treatments, we could provide cardiologists with more information when choosing different treatment strategies.

Through the early detection of microvascular changes in the retina, we might be able to speculate regarding the overall vascular condition of patients. Thus, we can also suggest that the OCTA scanning of the retina in patients with ICAS be used as a screening method alongside carotid Doppler US.

We acknowledge several limitations to this study.

First, the small sample size of 98 subjects is a limitation of this study. Our research included only four patients aged over 70, as well as four subjects aged over 69. It is accepted that about 5–10% of all individuals aged 70 years or older are affected by carotid stenosis with ≥50% luminal narrowing [1,45]. Our research did not find patients with severe carotid artery stenosis, only mild ICAS and moderate ICAS. We did not compare the OCTA retinal microvasculature between ipsilateral and contralateral eyes. During stenosis, ipsilateral circulation may rely on contralateral circulation for support as a compensatory mechanism, which may explain the insignificant difference between the contralateral and ipsilateral retinal microvasculature. Given the small sample size of our study, future research with larger sample sizes is needed.

Second, no analysis was performed regarding the medication used by the patients, which could have had a potential effect on OCTA results. Most patients were using multiple medications: antihypertensive drugs, statins and even anticoagulants.

Third, no retinal functional tests that may have revealed a link between retinal functions and anatomic alterations were performed. Several other methods can be used to evaluate ocular vessels, such as color Doppler imaging and fundus photography [16,17,20]. Color Doppler imaging can assess the velocity of blood flow in the ocular vessels, in the central retinal artery, in the posterior ciliary artery and in the ophthalmic artery. Fundus photography is another method used to evaluate ocular changes in patients with ICAS, as it can measure the retinal vascular caliber.

## 5. Conclusions

With the help of OCTA, we found that retinal and choroidal microvasculature changes were closely related to carotid stenosis.

Retinal microvascular changes are a prominent outcome of internal carotid disease and even mild stenosis can lead to alterations in the retinal microvascular bed which could be detected by OCTA.

Since the majority of our patients consisted of subjects with mild and moderate stenosis (22), we can conclude that OCTA could be used to recognize mild disease. OCTA may be used as a tool for monitoring the microvascular changes associated with carotid stenosis and provide an insight concerning the overall vascular condition of the patient, since OCTA produces subjective data on vessel density and flow.

Furthermore, we demonstrated that OCTA can be an efficient and non-invasive measurement method for detecting retinal microvascular changes in patients with ICAS. The retinal microvascular changes assessed by OCTA may serve as a novel biomarker reflecting the severity of ICAS and potentially identifying ICAS.

Therefore, we propose that OCTA can be used as a screening method in patients with ICAS, complementary to carotid Doppler US at the same time during medical evaluation.

Further research with larger patient groups, additional imaging of the retina and systemic parameters could shed more light on this issue.

Ophthalmologic symptoms and findings may often be the only manifestation in patients with carotid atherosclerotic disease. It is important for ophthalmologists and cardiologists to be familiar with these symptoms and to prevent them. Hence, high-risk patients for an ischemic stroke can be identified and be referred for early intervention or a change in medication.

## Figures and Tables

**Figure 1 diagnostics-15-01393-f001:**
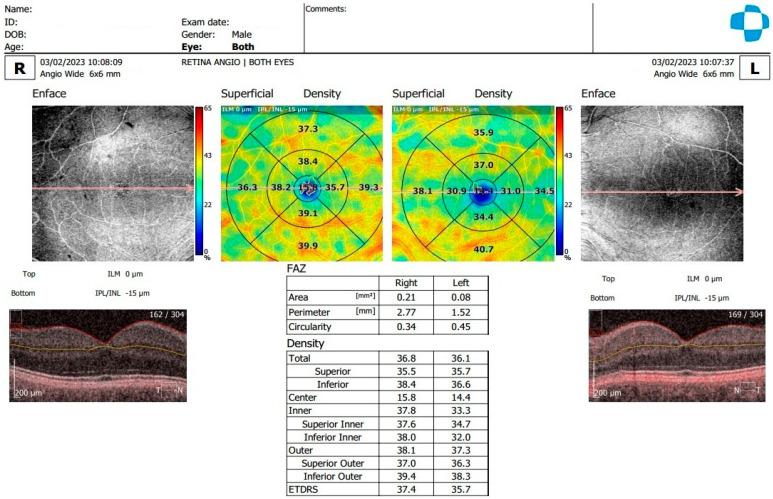
En face OCTA of the superficial vascular plexus with calculated FAZ area, comparing the right eye and left eye. Density is calculated and divided into total, center, inner and outer density. The ETDRS grid is also available, comparing right eye and left eye. OCTA shows that the superficial plexus extends from the inner limiting membrane (top) (0 µm) to the inner plexiform layer/inner nuclear layer (bottom) (IPL/INL-15 µm).

**Figure 2 diagnostics-15-01393-f002:**
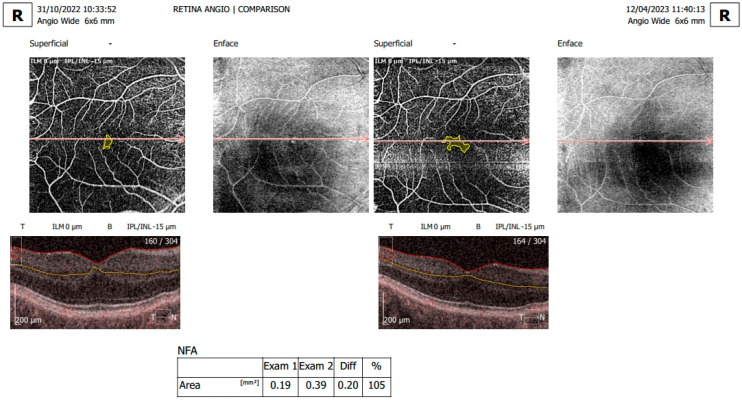
En face OCTA of the superficial vascular plexus with calculated NFA, comparing right eye examination number 1 and examination number 2. The difference between the examinations is measured in mm^2^ and expressed as a percentage.

**Figure 3 diagnostics-15-01393-f003:**
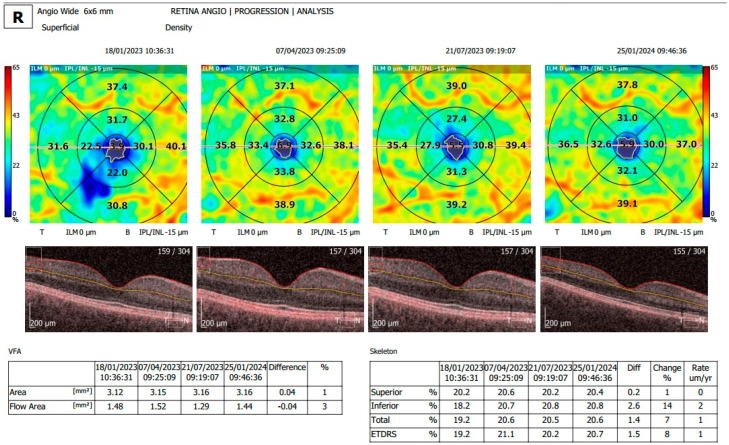
Retina Angio progression analysis program comparing four scans in the right eye. VFA and flow area were measured and compared. The superior zone, inferior zone, total skeleton and ETDRS grid are shown in a table in the right bottom corner of the figure, with the calculated difference and change rate in µm/year between the four scans provided.

**Figure 4 diagnostics-15-01393-f004:**
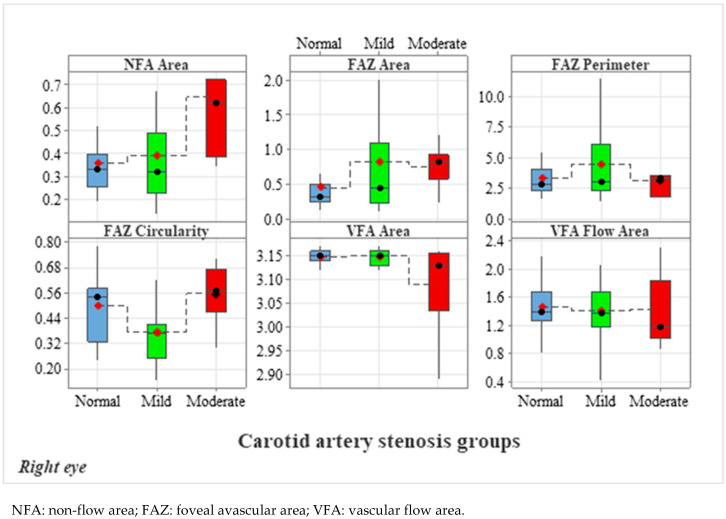
Right eye calculations of OCTA parameters (FAZ, NFA and VFA) and data distribution according to each carotid stenosis group. The normal group (no stenosis) is displayed in blue, the mild stenosis group is displayed in green and the moderate stenosis group is displayed in red.

**Figure 5 diagnostics-15-01393-f005:**
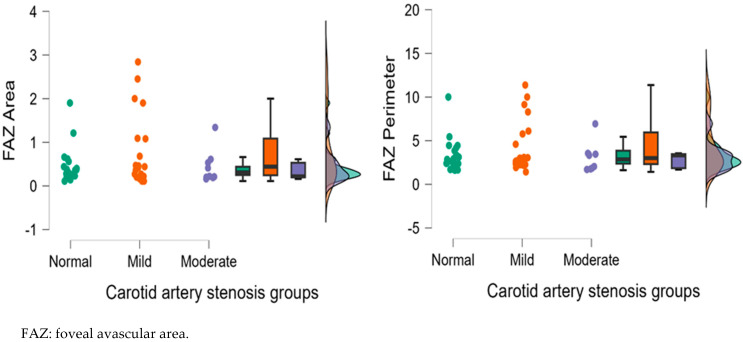
FAZ (area and perimeter) calculations in the right eye group according to each carotid stenosis group. Distribution of data as a scatter plot, boxplot and normal curve.

**Figure 6 diagnostics-15-01393-f006:**
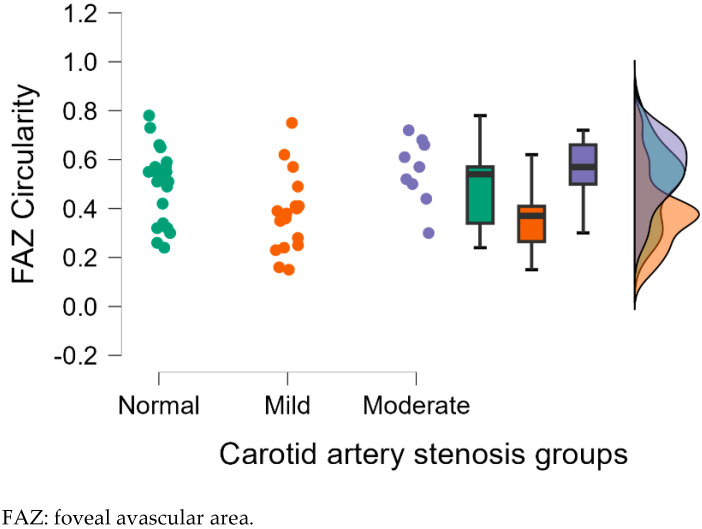
FAZ circularity calculations in the right eye group according to each carotid stenosis group. Distribution of data as a scatter plot, boxplot and normal curve.

**Figure 7 diagnostics-15-01393-f007:**
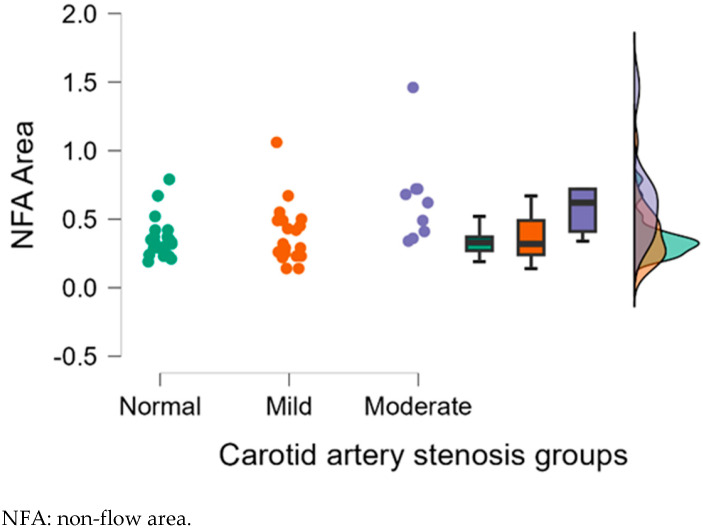
Right eye NFA values and data distribution as a scatter plot, boxplot and normal curve according to each carotid stenosis group.

**Figure 8 diagnostics-15-01393-f008:**
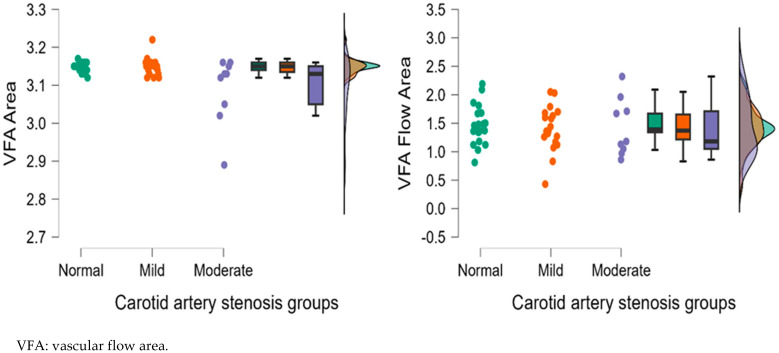
Right eye VFA (area and flow area) calculations according to each carotid stenosis group. Distribution of data as a scatter plot, boxplot and normal curve.

**Figure 9 diagnostics-15-01393-f009:**
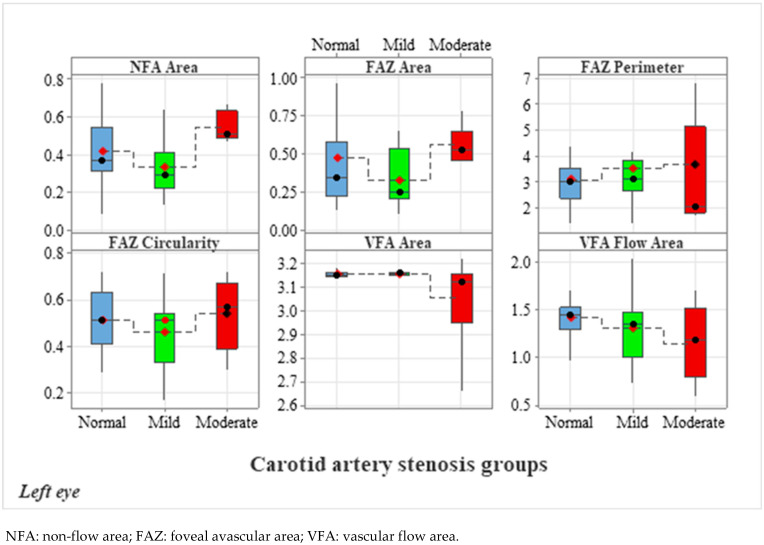
Left eye calculations of OCTA parameters (FAZ, NFA and VFA) and distribution of data according to each carotid stenosis group. Normal group (no stenosis) is displayed in blue, mild stenosis group is displayed in green and moderate stenosis group is displayed in red.

**Figure 10 diagnostics-15-01393-f010:**
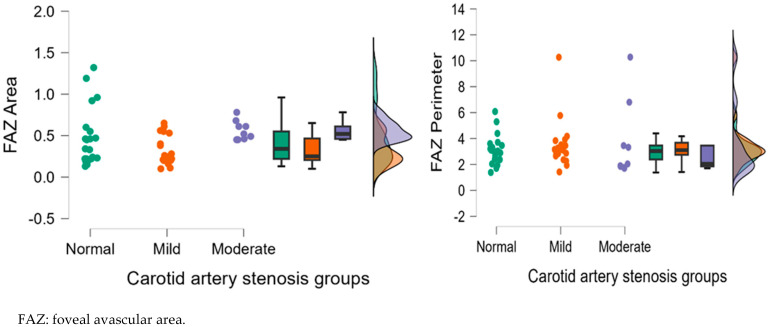
FAZ (area and perimeter) calculations in the left eye group according to each carotid stenosis group. Distribution of data as a scatter plot, boxplot and normal curve.

**Figure 11 diagnostics-15-01393-f011:**
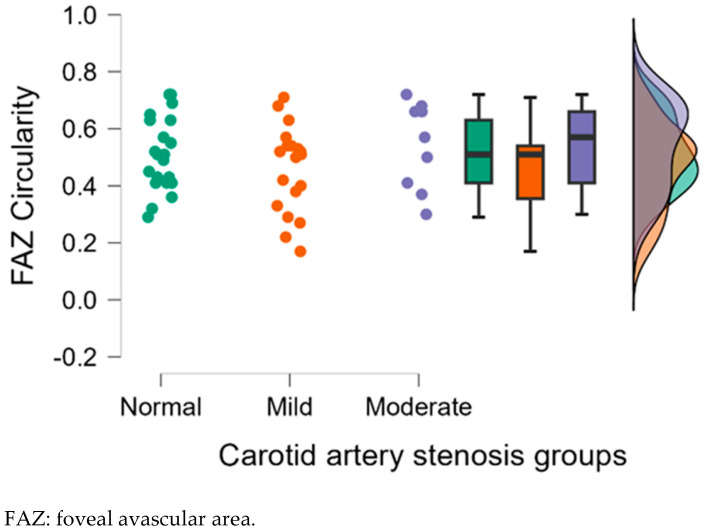
FAZ circularity calculations in the left eye group according to each carotid stenosis group. Distribution of data as a scatter plot, boxplot and normal curve.

**Figure 12 diagnostics-15-01393-f012:**
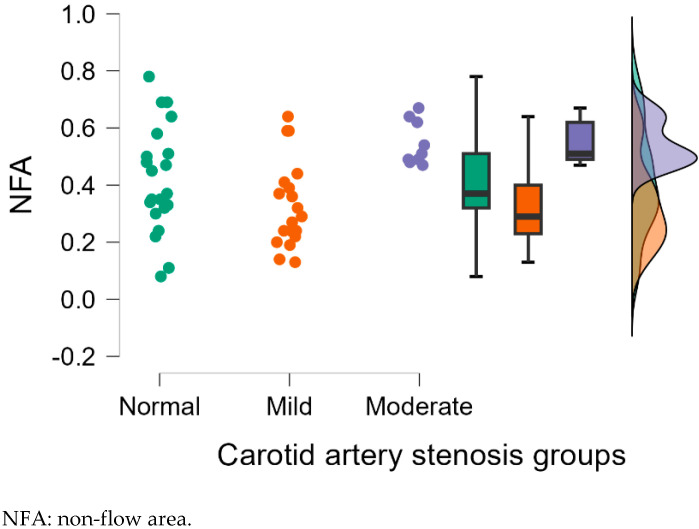
Left eye NFA values and distribution as a scatter plot, boxplot and normal curve according to each carotid stenosis group.

**Figure 13 diagnostics-15-01393-f013:**
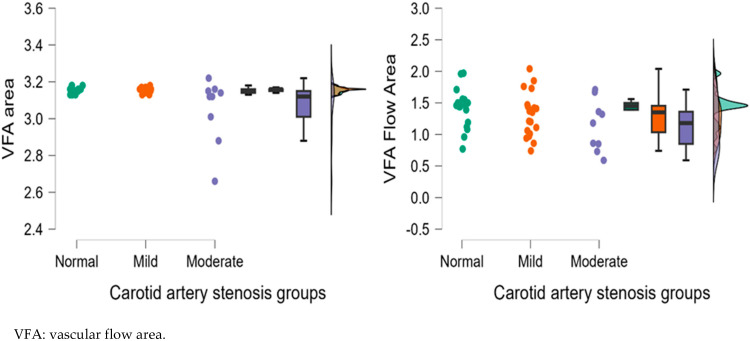
Left VFA (area and flow area) distribution of data as a scatter plot, boxplot and normal curve according to each carotid stenosis group.

**Figure 14 diagnostics-15-01393-f014:**
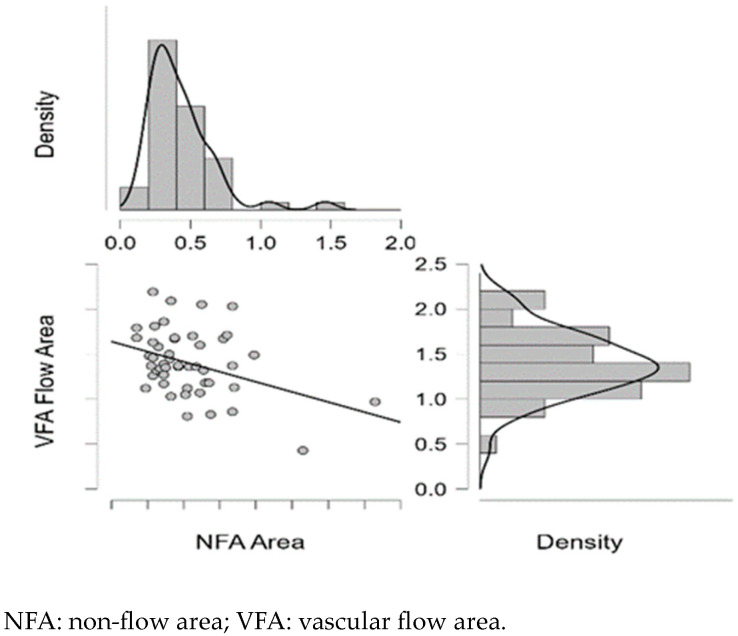
Right eye QQ plot for right NFA and VFA.

**Figure 15 diagnostics-15-01393-f015:**
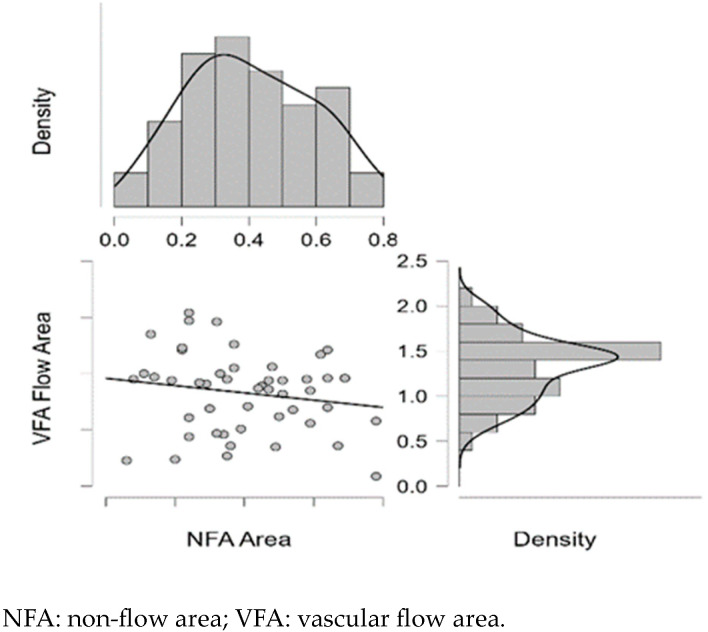
Left eye QQ plot for left NFA and VFA.

**Table 1 diagnostics-15-01393-t001:** Demographic parameters.

Demographic Parameters	Carotid Artery Stenosis Groups			*p*-Value *
	Normal (N = 21)	Mild (N = 19)	Moderate (N = 9)	
Gender				0.614
Female	10 (47.6%)	7 (36.8%)	5 (55.6%)	
Male	11 (52.4%)	12 (63.2%)	4 (44.4%)	
Smoker?				0.002 *
YES	4 (19.1%)	7 (36.8%)	8 (88.9%)	
NO	17 (80.9%)	12 (63.2%)	1 (1.1%)	
Age ** (years)	54.4 ± 10.0	60.9 ± 8.2	58.2 ± 13.1	0.057

* ꭓ2 test for gender and smoking habit. Kruskal–Wallis test for age. ** Age calculated as M ± SD. M: mean deviation; SD: standard deviation; n: number of subjects per group.

**Table 2 diagnostics-15-01393-t002:** Correlation between right eye OCTA parameters according to the Kruskal–Wallis test and Dunn’s post hoc comparisons.

Right Eye OCTA Parameters	Carotid Artery Stenosis Groups			Kruskal–Wallis Test		
				Dunn’s Post Hoc Comparisons		
	Normal (n = 21)	Mild (n = 19)	Moderate (n = 9)	Normal-Mild *p*-Value	Normal-Moderate *p*-Value	Mild-Moderate *p*-Value
	**M ± SD**	**M ± SD**	**M ± SD**			
NFA	0.36 ± 0.15	0.39 ± 0.22	0.64 ± 0.34	0.754	0.004 **	0.011 *
FAZ area	0.45 ± 0.41	0.82 ± 0.85	0.75 ± 0.28	0.191	0.016 *	0.178
FAZ Perimeter	3.31 ± 1.84	4.46 ± 3.07	3.11 ± 1.64	0.447	0.820	0.413
FAZ Circularity	0.50 ± 0.15	0.38 ± 0.15	0.56 ± 0.13	0.025 *	0.325	0.006 **
VFA	3.15 ± 0.01	3.15 ± 0.02	3.09 ± 0.09	0.772	0.037 *	0.068
VFA	1.46 ± 0.34	1.41 ± 0.40	1.43 ± 0.51	0.729	0.641	0.852

* *p* < 0.05, ** *p* < 0.01. NFA: non-flow area; FAZ: foveal avascular area; VFA: vascular flow area.

**Table 3 diagnostics-15-01393-t003:** Correlation between left eye OCTA parameters according to the Kruskal–Wallis test and Dunn’s post hoc comparisons.

Left Eye OCTA Parameters	Carotid Artery Stenosis Groups			Kruskal–Wallis Test		
				Dunn’s Post Hoc Comparisons		
	Normal (n = 21)	Mild (n = 19)	Moderate (n = 9)	Normal-Mild *p*-Value	Normal-Moderate *p*-Value	Mild-Moderate *p*-Value
	M ± SD	M ± SD	M ± SD			
NFA	0.42 ± 0.19	0.33 ± 0.15	0.55 ± 0.08	0.101	0.045 *	0.001 *
FAZ Area	0.47 ± 0.35	0.32 ± 0.18	0.56 ± 0.12	0.298	0.054	0.007 *
FAZ Perimeter	3.08 ± 1.13	3.50 ± 1.89	3.68 ± 2.96	0.565	0.626	0.352
FAZ Circularity	0.51 ± 0.13	0.46 ± 0.15	0.54 ± 0.15	0.379	0.572	0.213
VFA	3.15 ± 0.02	3.16 ± 0.01	3.05 ± 0.18	0.554	0.041 *	0.013 *
VFA	1.42 ± 0.29	1.31 ± 0.36	1.14 ± 0.41	0.161	0.029 *	0.294

* *p* < 0.05. NFA: non-flow area; FAZ: foveal avascular area; VFA: vascular flow area.

**Table 4 diagnostics-15-01393-t004:** Pearson and Spearman correlation among right eye OCTA parameters and right carotid Doppler US parameters.

Right Carotid Doppler Parameters–Right Eye OCTA Parameters			Pearson		Spearman	
			r	*p*	rho	*p*
ICA PSV	-	NFA	0.180	0.216	0.068	0.641
ICA PSV	-	VFA	−0.260	0.072	−0.286 *	0.046
ICA EDV	-	NFA	0.365 *	0.010	−0.089	0.543
ICA EDV	-	VFA	−0.086	0.555	0.052	0.723
ECA PSV	-	NFA	0.219	0.131	0.130	0.373
ECA PSV	-	VFA	−0.288 *	0.045	−0.317 *	0.026
ECA EDV	-	NFA	−0.198	0.173	−0.15	0.305
ECA EDV	-	VFA	−0.057	0.697	−0.058	0.692
CCA PSV	-	NFA	−0.256	0.076	−0.354 *	0.012
CCA PSV	-	VFA	0.112	0.443	0.067	0.647
CCA EDV	-	NFA	−0.262	0.069	−0.292 *	0.042
CCA EDV	-	VFA	0.108	0.461	−0.053	0.718
VA PSV	-	NFA	0.195	0.179	0.074	0.612
VA PSV	-	VFA	−0.327 *	0.022	−0.379 **	0.007
VA EDV	-	NFA	−0.087	0.550	−0.190	0.192
VA EDV	-	VFA	−0.031	0.831	−0.063	0.669
NFA	-	VFA	−0.374 **	0.008	−0.288 *	0.045

* *p* < 0.05, ** *p* < 0.01. NFA: non-flow area; FAZ: foveal avascular area; VFA: vascular flow area; PSV: peak systolic velocity; EDV: end-diastolic velocity; ICA: internal carotid artery; ECA: external carotid artery; CCA: common carotid artery; VA: vertebral artery.

**Table 5 diagnostics-15-01393-t005:** Pearson and Spearman correlation among left eye OCTA parameters and left carotid Doppler US parameters.

Left Carotid Doppler Parameters–Left Eye OCTA Parameters			Pearson		Spearman	
			r	*p*	rho	*p*
ICA PSV	-	NFA	−0.351 *	0.013	−0.313 *	0.028
ICA PSV	-	VFA	−0.227	0.117	−0.140	0.336
ICA EDV	-	NFA	0.019	0.899	−0.003	0.983
ICA EDV	-	VFA	−0.421 **	0.003	−0.314 *	0.028
ECA PSV	-	NFA	−0.412 **	0.003	−0.457 ***	0.001
ECA PSV	-	VFA	0.007	0.962	0.041	0.778
ECA EDV	-	NFA	−0.064	0.664	−0.080	0.585
ECA EDV	-	VFA	−0.008	0.954	0.030	0.836
CCA PSV	-	NFA	−0.288 *	0.045	−0.339 *	0.017
CCA PSV	-	VFA	0.094	0.521	0.109	0.455
CCA EDV	-	NFA	−0.404 **	0.004	−0.417 *	0.003
CCA EDV	-	VFA	0.164	0.261	0.092	0.530
VA PSV	-	NFA	−0.043	0.770	−0.039	0.789
VA PSV	-	VFA	−0.169	0.246	−0.179	0.218
VA EDV	-	NFA	−0.004	0.978	−0.037	0.801
VA EDV	-	VFA	−0.033	0.822	−0.033	0.821
NFA	-	VFA	−0.176	0.228	−0.169	0.245

* *p* < 0.05, ** *p* < 0.01, *** *p* < 0.001. NFA: non-flow area; FAZ: foveal avascular area; VFA: vascular flow area; PSV: peak systolic velocity; EDV: end-diastolic velocity; ICA: internal carotid artery; ECA: external carotid artery; CCA: common carotid artery; VA: vertebral artery.

**Table 6 diagnostics-15-01393-t006:** Right eye Shapiro–Wilk test for multivariate normality.

Shapiro–Wilk	*p*
0.928	<0.001

**Table 7 diagnostics-15-01393-t007:** Left eye Shapiro–Wilk test for multivariate normality.

Shapiro–Wilk	*p*
0.958	<0.001

## Data Availability

The data that support the findings of the present study are available from the corresponding author upon reasonable request.

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
