# Peer review of "The Relevance of Optical Coherence Tomography Angiography in Screening and Monitoring Hypertensive Patients with Carotid Artery Stenosis"

_diagnostics, 2025, doi:10.3390/diagnostics15111393_

Round 1
Reviewer 1 Report
Comments and Suggestions for Authors
This study utilized optical coherence tomography angiography (OCTA) and Carotid Doppler Ultrasonography to image both eyes of 43 recruited patients, including 21 healthy individuals, 19 with mild carotid artery stenosis, and 3 with moderate stenosis. The study assessed the statistical differences in OCTA images among the three groups and evaluated the correlation between carotid artery stenosis and retinal microvascular changes, highlighting the potential of OCTA as a screening tool. Some improvements need to be made.
Abstract:
- To improve clarity for readers, please revise the Background section in the abstract. First, clearly state why this study is necessary and what the clinic needs, followed by the research objectives.
- In the Methods section, specify the statistical methods used and the parameters calculated, such as area and blood flow velocity.
- The description of the Results section needs to be clearer. For "in patients with right ICAS," do you mean the right eye group? you analyzed both the left and right eye groups, correct? What about the results for patients with left ICAS?
- There is an issue with the Results section description:
"Statistically significant results were found for internal carotid artery (ICA) end-diastolic velocity (EDV) and NFA Area (r=0.440; p=0.003) in the right eye group."
You mentioned a correlation coefficient, so I assume you meant to indicate a correlation between velocity and area, rather than a statistical difference. Please clarify. - In the sentence:
"Correlations were also found between external carotid artery (ECA) PSV and NFA Area (r=0.363; p=0.017), (rho=0.389; p=0.008), and for common carotid artery (CCA) EDV and NFA Flow Area (r=-0.411; p=0.006), (rho=-0.427; p=0.004)."
Is this describing the left or right eye group? Please specify. - The Conclusions section is too wordy. Please provide a more concise summary of the study's significance and if the results achieved your aims or verified your hypothesis.
Introduction :
- Improve the Introduction section to enhance its logical flow and conciseness. Avoid redundant or repetitive statements to improve readability and impact.
- For example, the sentences:
"Microvascular alterations are thoroughly associated with cardiovascular changes [3, 4]." (Third paragraph)
"Microvascular abnormalities are demonstrated to be important contributors to cardiovascular disease [2, 8, 10]." (Seventh paragraph)
are very similar and should be streamlined for clarity. - the following passage:
"Thus, the Optical Coherence Tomography Angiography (OCTA) retinal imaging technique can be used to provide a detailed analysis of retinal vascular parameters. OCTA is a novel, dye-free, non-invasive, and efficient method for qualitative and quantitative assessment of retinal vessels. OCTA has changed the way of quantifying and visualizing the retinal microvasculature." The logic needs to be improved. - It does not make sense to include the following sentence at the end of the Introduction:
"For routine screening of subclinical atherosclerosis of the carotid artery in the general population, availability, ease of measurement, and cost are essential when choosing the proper imaging methodology [23, 28]."
Consider moving it elsewhere or revising its relevance in the Introduction.
Methods:
Section 2.1: Improve the logical flow and descriptions.
- For example, the statement: "All ultrasound imaging was performed by the same cardiologist."(third paragraph)
is redundant with: "US was performed by the same cardiologist." (14th paragraph). Remove redundancy. - Define abbreviations(e.g., FAZ, NFA) in full when first mentioned in the main text.
Section 2.2:
- Figures 1, 2, and 3: Please rearrange them and provide a clear description in the text. Ensure that the figures explicitly present the information you want readers to understand, rather than just displaying raw outputs from software.
Results:
- Provide some descriptions for Table 1, Figures 4, 5, and 6 within the main text and figure captions.
- Figures 8, 9, and 10 do not provide significantly different information from Figure 7; they merely present scatter plots and box plots. Additionally, the image quality and aspect ratio are suboptimal.
-
- Consider moving Figures 8, 9, and 10 to Supplementary Materials and improving their quality.
- Similar issues exist with other figures in the Results section
3. Please ensure consistent formatting between Table 2 and Table 3. The current arrangement is confusing because the column of the normal group in Table 3 differs from the normal group in Table 2. Standardizing the presentation will improve clarity and avoid confusion for the readers.
4. The moderate group includes only three cases, which may not provide statistically reliable results. For example, in Table 2, the NFA area shows a large standard deviation for the normal and mild groups, whereas the moderate group has a small standard deviation despite its small sample size. Additionally, the mean values for the normal and moderate groups are similar, yet a significant difference is reported.
Comments on the Quality of English LanguageThe logical flow and clarity of the manuscript need to be improved.
Author Response
Dear reviewer,
Thank you for your guidance and suggestions!
I hope my changes are according to your request.
- I have revised the Abstract, I have clearly specified the right eye and the left eye group.
- The Introduction is according to your request – I was not a logical flow as you mentioned.
- Methods section – I meant to underline that having the same Cardiologist will make de US scanning more accurate. I have changed the text. Abbreviations are corrected and placed in order.
- Results- the figures have a description.
- Please ensure consistent formatting between Table 2 and Table 3: thank you for that suggestion as well! I used the same formatting in the two tables this time.
As regarding to probably the main issue with my manuscript:
The moderate group includes only three cases, which may not provide statistically reliable results. For example, in Table 2, the NFA area shows a large standard deviation for the normal and mild groups, whereas the moderate group has a small standard deviation despite its small sample size. Additionally, the mean values for the normal and moderate groups are similar, yet a significant difference is reported. – In this short time I have recruited more patients (this resulted in a total number of 9 patients with moderate stenosis) and I worked with a statistician in order to analyze all data, so almost all data has changed. I agree it is still a small number of patients, but the data is correct and I think interesting. So thank you for that suggestion! I think the study should be more relevant this time.
I have contacted the author services to help with English revision but I wanted to know your professional opinion on the manuscript first.
With respect,
Dr. Barca Irina
Reviewer 2 Report
Comments and Suggestions for Authors
The early detection and evaluation ICAS using OCTA is an interesting clinical topic. However, I have some major concerns on the manuscript.
- This topic has been investigated in some previous studies (e.g.,10.3389/fnins.2024.1361413). It will be important to comprehensively review the literature and compare the results with others'.
- Regarding the background, it is essential to introduce the relationship between different parts of cerebral circulation, e.g., extra- and intracranial arteries, on the microvascular level (Refer: 10.3389/fphys.2023.1085871).
- In the abstract, the methods part is more about scientific hypotheses. Please add the details of methodology.
- The methodology need some major revision. Some parts like the normality tests can be simplified. The correlations between different OCT parameters need to be provided. Figures 4-6 are not necessary. Instead, it will be helpful to provide a table that covers all the basic information of patients, with essential tests. I suggest to add a statistician to review the methodology and provide some suggestions since fundamental changes are needed.
Author Response
Dear reviewer,
Thank you for your guidance and suggestions!
I hope my changes are according to your request.
- I have revised the studies you indicated. Thank you! They are very good and appreciated. My purpose was not to assume that my work was never done before, but to present to you and to readers of your Journal my own data on a subject I find interesting.
- Regarding the background, it is essential to introduce the relationship between different parts of cerebral circulation, e.g., extra- and intracranial arteries, on the microvascular level – I have introduced this part in my article and I actually think this time my manuscript (thanks to your suggestion) has a complete information.
- In the abstract, the methods part is more about scientific hypotheses. Please add the details of methodology. I took care of that.
- The methodology need some major revision. Some parts like the normality tests can be simplified. The correlations between different OCT parameters need to be provided. Figures 4-6 are not necessary. Instead, it will be helpful to provide a table that covers all the basic information of patients, with essential tests. I suggest to add a statistician to review the methodology and provide some suggestions since fundamental changes are needed. – I have change the whole section here according to your request. The normality tests are simplified and most figures are in the Supplementary materials. The correlations between different OCT parameters are provided and I have added some figures as well.
As regarding to probably the main issue with my manuscript:
The moderate group included only three cases, which may not provide statistically reliable results– In this short time (two months since I have summited my article) I have recruited more patients (this resulted in a total number of 9 patients with moderate stenosis) and I worked with a statistician in order to analyze all data, so almost all data has changed. I agree it is still a small number of patients, but the data is correct and I think interesting. I think the study should be more relevant this time.
I have contacted the author services to help with English revision but I wanted to know your professional opinion on the manuscript first.
With respect,
Dr. Barca Irina
Reviewer 3 Report
Comments and Suggestions for Authors
In this article, the authors analyzed and compared vascular measurements
provided by optical coherence tomography angiography (OCTA) and Carotid Doppler
ultrasonography (US) aimed at examining and correlating retinal microvascular
changes in patients with hypertensive retinopathy, dyslipidemia and internal carotid
artery stenosis (ICAS). The results show that OCTA can be an efficient, cost-effective
and non-invasive measurement method for detecting early-stage ICAS, therefore it
can be used to reduce the incidence of cerebral and cardiovascular complications at
later stages. OCTA monitoring could lead to an increased efficiency in screening and
diagnosing patients with IACS.
General, the manuscript exhibits a commendable presentation and interesting
ideas. However, certain revisions are necessary before it can be considered for
publication. The following comments and suggestions have been provided.
On the science of the paper:
1. There are several instances of incorrect paragraph spacing in the text, such as
between the fourth and fifth paragraphs and between the eighth and ninth
paragraphs of the first section Introduction on page 2. These are just two
examples. The author needs to proofread the entire text to revise the others.
2. On page 3, the authors mentioned that ‘patients with mild stenosis are usually
asymptomatic’. In the absence of clinical signs and symptoms, how do you
determine that an average person has mild stenosis so that you can test them
with OCTA or Carotid Doppler US?
3. On page 4, the authors mentioned that ‘we selected diagnosed patients with
chronic hypertension, secondary dyslipidemia and chronic hypertensive
retinopathy.’ And ‘Patients were enrolled irrespective of current treatment for
hypertension and dyslipidemia.’ Does the current treatment for hypertension and
dyslipidemia have an effect on the results of the OCTA test, and do the
medications the participants were taking as part of the treatment have an effect
on the microvasculature of the retina?
4. On page 5, the authors mentioned that ‘Based on fundoscopy we included in the
study subjects with grade I and II of hypertensive retinopathy.’ Why were subjects
with grade I and II hypertensive retinopathy included in the study while
participants with grade III and IV hypertensive retinopathy were excluded? What
was the basis for excluding these participants? Did any of these excluded
participants have retinopathy due to internal carotid artery stenosis? If these
participants had not been excluded, but had been included, would there have
been any impact on the results of the study?
5. There are also several instances of incorrect punctuation in the text, such as an
extra period in ‘Figure 3. Retina Angio progression analysis program comparing 4
scans in the right eye..’ on page 8 and a missing period in the footnote to table 1
on page 9. These are just two examples. The author needs to proofread the
entire text to revise the others.
6. Authors should check and harmonize footnote information. For example, ‘* p <
0.05, ** p < 0.01, *** p < 0.001.’ in Table 4 on page 16 and ‘* p < .05, ** p < .01,
*** p < .001.’ in Table 5 on page 17. In addition, authors should check the text for
semantic fluency, for example, on page 24 ‘The Kruskal-Wallis test and Dunn's
post-hoc test were applied to determine if there are statistically significant
differences between normal, mild and moderate stenosis groups. Kruskal-Wallis
test.’ There is extra sentence ‘Kruskal-Wallis test.’ at the end of the sentence.
The English could be improved
Author Response
Dear reviewer,
Thank you for your guidance and suggestions!
I hope my changes are according to your request.
General, the manuscript exhibits a commendable presentation and interesting ideas – Thank you!
However, certain revisions are necessary before it can be considered for publication – Yes, I think your suggestion is perfect.
The following comments and suggestions have been provided – Thank you! I will write my replies here:
- There are several instances of incorrect paragraph spacing in the text- I have revised the entire text.
- Patients with mild stenosis are usually asymptomatic’ - Though severe ICAS may lead to clinical visual signs and symptoms, patients with mild stenosis diagnosed on Carotid Doppler US are usually asymptomatic. I hope I have made it clear this time that I was talking about visual symptoms.
- Does the current treatment for hypertension and dyslipidemia have an effect on the results of the OCTA test, and do the medications the participants were taking as part of the treatment have an effect on the microvasculature of the retina?
I have specified our choice in the text: The aim of our research was to enroll subjects irrespective of current treatment for hypertension and dyslipidemia in order to study the correlation between hypertensive retinopathy and ICAS on all patients. We did not aim to study the effect of medication on the microvasculature of the retina.
- Why were subjects with grade I and II hypertensive retinopathy included in the study while participants with grade III and IV hypertensive retinopathy were excluded? Patients with Grade III and IV were not excluded. Our research just did not find patients with stage III and IV because these stages are not chronic patients. I have made a mention in the article.
Based on fundoscopy we included in the study subjects with grade I and II of hypertensive retinopathy according to the Keith–Wagener–Barker and the Mitchell–Wong classification system [28]. Our research did not find participants with grade III and IV of hypertensive retinopathy. Based on Carotid Doppler US evaluation and on Society of Radiologists in Ultrasound consensus patients were classified and all degrees of ICAS were included in our study.
- And 6. – I have corrected the text.
I have contacted the author services to help with English revision but I wanted to know your professional opinion on the manuscript first.
With respect,
Dr. Barca Irina
Reviewer 4 Report
Comments and Suggestions for Authors
This is a well-organized and relevant study that investigates the potential of OCTA as a non-invasive biomarker for internal carotid artery stenosis (ICAS) in dyslipidemic patients with hypertension. The association between retinal microvascular changes and grades of ICAS is relevant and timely. The authors have a thought-provoking hypothesis based on evidence from a well-characterized prospective cohort study.
The introduction, though helpful, may be more concisely structured. A number of key concepts like the systemic importance of retinal microcirculation and the utility of OCTA are repeated multiple times. An even more synoptic and thematically structured introduction would be more readable without a sacrifice of scientific context.
Though findings are interesting, only 3 patients with moderate stenosis undermines the strength of statistical comparison by a huge degree. Mention has been made of this drawback in the discussion section, yet greater emphasis has to be brought about on undermining the stability of conclusions, in particular on ICAS that was moderate.
The discussion would be enhanced by more robust comparison with similar studies that have applied OCTA in systemic cardiovascular disease. Although the authors mention cardiovascular disease in general terms, more precise mention of previously published OCTA-based studies in systemic disease would enhance contextualization of the findings.
I suggest including two additional references in the discussion to contextualize OCTA's place more robustly in systemic vascular pathology. These articles have already studied OCTA in patients with systemic cardiovascular disease and would make the study more impactful: Rinaldi M, Chiosi F, Passaro ML, et al. Resistive index of central retinal artery, aortic arterial stiffness and OCTA correlated parameters in the early stage of Fabry disease. Sci Rep. 2024 Oct 14;14(1):24047. doi: 10.1038/s41598-024-74146-5.
Chiosi F, Campagna G, Rinaldi M, et al. Optical Coherence Tomography Angiography Analysis of Vessel Density Indices in Early Post-COVID-19 Patients. Front Med (Lausanne). 2022 Jun 28;9:927121. doi: 10.3389/fmed.2022.927121.
There may be an explanation of why some OCTA parameters (FAZ, NFA, VFA) have been selected, which might help to inform on why they are expected to change with ICAS and how different they are compared to other known microvascular indices.
The article would benefit from professional editing of language to clarify the occasional grammatical mistake and improve clarity. For example, terms like "subjective data" can be changed to "objective data" when referring to quantifiable OCTA data.
All abbreviations (e.g., NFA, VFA, FAZ) need to be defined on their first use in the abstract as well as body text!
There are certain figures, particularly Figures 13–20, which can be more clearly described in their legends to guide the reader through the interpretation.
A valuable contribution to the growing body of non-invasive vascular imaging. After above points, particularly the ones related to the issue of the limitations of sample size and integrated literature coverage, are considered, the manuscript will be eligible for publication
Author Response
Dear reviewer,
Thank you for your guidance and suggestions!
I hope my changes are according to your request.
General, the manuscript exhibits a commendable presentation and interesting ideas – Thank you!
However, certain revisions are necessary before it can be considered for publication – Yes, I think your suggestion is perfect.
The following comments and suggestions have been provided – Thank you! I will write my replies here:
- This is a well-organized and relevant study that investigates the potential of OCTA as a non-invasive biomarker for internal carotid artery stenosis (ICAS) in dyslipidemic patients with hypertension. The association between retinal microvascular changes and grades of ICAS is relevant and timely. The authors have a thought-provoking hypothesis based on evidence from a well-characterized prospective cohort study. Thank you! Your kind words are appreciated!
- An even more synoptic and thematically structured introduction would be more readable without a sacrifice of scientific context. I have structured the introduction.
- Though findings are interesting, only 3 patients with moderate stenosis undermines the strength of statistical comparison by a huge degree. -- In this short time I have recruited more patients (this resulted in a total number of 9 patients with moderate stenosis) and I worked with a statistician in order to analyze all data, so almost all data has changed. I agree it is still a small number of patients, but the data is correct and I think interesting. So thank you for that suggestion! I think the study should be more relevant this time.
- The discussion would be enhanced by more robust comparison with similar studies that have applied OCTA in systemic cardiovascular disease. I have studied and referenced other studies thw two studies mentioned by you and others (I suggest including two additional references in the discussion to contextualize OCTA's place more robustly in systemic vascular pathology). Your suggestion was very usefull, thank you!
- There may be an explanation of why some OCTA parameters (FAZ, NFA, VFA) have been selected. I have added teh explination in the manuscript: page 10:
The OCTA software provides assessment of the FAZ, NFA, VFA areas and the retinal vascular density and skeleton. Our research included only FAZ, NFA and VFA. There areas are detected and measured using the semi-auto area tool of the software thus the results prove to be more objective and accurate. The A.I. system of the OCTA has a direct impact on the accuracy of the clinical assessment and the assessment of the status of areas of pathology in the retina. This level of detection is more accurate and this results in more detailed screening. A.I. segmentation is important for follow-up examinations, bringing a more accurate diagnosis when analyzing pathology over time.
- All abbreviations (e.g., NFA, VFA, FAZ) need to be defined: I have defined them
- Figures 13–20, which can be more clearly described: I have added desription and clearer Figures. Almost all of them are changed in order to be more clearer.
I have contacted the author services to help with English revision but I wanted to know your professional opinion on the manuscript first.
With respect,
Dr. Barca Irina
Round 2
Reviewer 1 Report
Comments and Suggestions for Authors
The revised manuscript has improved in terms of logical flow and clarity. Additional data have been included, which strengthens the overall conclusions. I believe the manuscript is now close to being acceptable for publication.
There are still a few minor issues:
- Page 10: Please carefully check the sentence on Page 10: “Table 1, Figure 4, 5 and 6 show results of …” .
- Figure Captions (Figures 4–8): Please provide more informative captions for Figures 4, 5, 6, 7, and 8. A reader should be able to understand the key message of each figure based on its caption alone.
One suggestion: The current version contains track changes, which significantly hinder readability. While this is not a major issue for me, I recommend avoiding the submission of a tracked version for revision. Instead, consider highlighting the key changes in a clean version for clarity.
Author Response
Dear reviewer,
Thank you for you suggestion and remarks. I have revised the figures mentioned and I think this should be easier to understand for the journal`s readers.
Please find my manuscript attached. Changes are highlighted in red.
Sincerely,
Dr. Barca Irina

Reviewer 2 Report
Comments and Suggestions for Authors
Thanks for the update. My earlier comments have been well addressed. Please proofread and double check the format.
Author Response
Dear reviewer,
Thank you for your previous remarks and patience!
I sent my manuscript for English revision to the Author Services.
Please find my manuscript attached.
Sincerely,
Dr. Barca Irina

Reviewer 3 Report
Comments and Suggestions for Authors
All my questions have been revised by the author, and now I believe this paper can be published in Diagnostics journal.
Author Response
Dear reviewer,
Thank you for your previous suggestions and patience!
I sent my manuscript for English revision to the Author Services.
Please find my manuscript attached.
Sincerely,
Dr. Barca Irina
